# Telemedicine in Care of Sarcoma Patients beyond the COVID-19 Pandemic: Challenges and Opportunities

**DOI:** 10.3390/cancers15143700

**Published:** 2023-07-21

**Authors:** Christos Tsagkaris, Nikolaos Trygonis, Vasiliki Spyrou, Andreas Koulouris

**Affiliations:** 1European Student Think Tank, Public Health and Policy Working Group, 1058 DE Amsterdam, The Netherlands; chriss20x@gmail.com; 2Department of Orthopaedics, University Hospital of Heraklion, 70013 Heraklion, Greece; nikolaos.trygonis@gmail.com; 3Post Covid Department, Theme Female Health, Karolinska University Hospital, 14157 Stockholm, Sweden; vasiliki.spyrou21@gmail.com; 4Department of Oncology-Pathology, Karolinska Institute, 17176 Stockholm, Sweden; 5Thoracic Oncology Center, Theme Cancer, Karolinska University Hospital, 17177 Stockholm, Sweden; 6Faculty of Medicine, University of Crete, 70013 Heraklion, Greece

**Keywords:** Ewing sarcoma, coronavirus disease 2019 (COVID-19), gastrointestinal stromal tumor (GIST), osteosarcoma, sarcoma, soft tissue sarcoma (STS), telehealth, telemedicine, tele-oncology, young adult cancer

## Abstract

**Simple Summary:**

This review examines the utilization of telemedicine in sarcoma care during the COVID-19 pandemic. The authors conducted a systematic literature review of peer-reviewed studies following the Preferred Reporting Items for Systematic Reviews and Meta-Analyses (PRISMA) guidelines. The findings demonstrate the potential of telemedicine to improve clinical and psychological outcomes for sarcoma patients. This development builds on the progress of tele-oncology and telerehabilitation have shown significant progress over the past decade, with the COVID-19 outbreak exerting an accelerating effect on their adoption. Telehealth applications in sarcoma care include remote consultations, psychological support and virtual symptom assessment. Both healthcare providers and patients have reported satisfaction with telehealth services, at levels comparable to in-person visits. The study concludes that telehealth offers opportunities for tailored, individualized care and is likely to play a crucial role in sarcoma patient management in the post-COVID-19 era.

**Abstract:**

Background: The COVID-19 pandemic has created a challenging environment for sarcoma patients. Most oncology societies published guidelines or recommendations prioritizing sarcoma patients and established telehealth as an efficient method of approaching them. The aim of this review is the assessment of current evidence regarding the utilization of telemedicine in diagnosis, treatment modalities, telerehabilitation and satisfaction among sarcoma patients and healthcare providers (HP). Methods: This systematic review was carried out using the databases PubMed and Ovid MEDLINE according to the Preferred Reporting Items for Systematic Reviews and Meta-Analyses (PRISMA). Results: The application of telemedicine to the management of sarcoma has yielded improved clinical and psychological outcomes. Specifically, significant progress has been demonstrated in the areas of tele-oncology and telerehabilitation during the last decade, and the COVID-19 outbreak has accelerated this transition toward them. Telehealth has been proven efficient in a wide spectrum of applications from consultations on physical therapy and psychological support to virtual care symptom management. Both HP and patients reported satisfaction with telehealth services at levels comparable to in-person visits. Conclusions: Telehealth has already unveiled many opportunities in tailoring individualized care, and its role in the management of sarcoma patients has been established in the post-COVID-19 era, as well.

## 1. Introduction

### 1.1. History of Telemedicine in Oncology

Telemedicine is a compound word composed of tele- (meaning remote, from afar) and medicine. It is not certain when telemedicine was initiated as a concept and practice, but it has been reported that in 1906, Eindhoven, the inventor of the electrocardiogram, published a paper highlighting its potential telemedical applications [1]. Remote monitoring has been greatly advanced since the 1960s, when the increased need for space travel prompted action for remote monitoring, and treatment under the support and funding of space agencies such as the National Aeronautics and Space Administration (NASA) began performing physiologic monitoring over a distance [2].

The potential of telemedicine in cancer care has been recognized over the last few decades. On the one hand, telemedicine was seen as a way to bridge the gap between the increasing number of cancer patients and the limited number of cancer specialists. On the other hand, particular components of specialized cancer investigation and care are greatly suitable for a remote approach. An additional aspect of telemedicine with a major potential in oncology, a field that increasingly relies on the constant renewal of treatment guidelines, is the remote education of healthcare practitioners [3].

### 1.2. Telehealth and COVID-19

In early 2020, the scientific community and the general public were astonished to learn about the successful telemedical management of venous thrombosis during space flight [4]. Only a few months later, the spread of SARS-CoV-2 and the emergence of COVID-19 as a pandemic turned the tables and mandated the wide use of telemedicine as an urgent solution for the continuation of care of patients facing decreased access to hospitals. The growing inaccessibility of outpatient clinics and hence the limited access to cancer follow-up and screening prompted the integration of telemedicine in oncological settings [5,6].

During the pandemic, tele-oncological services were deployed in the entire spectrum of cancer care from primary prevention and screening to the follow-up of patients undergoing surgery, radiation therapy (RT), chemotherapy or immunotherapy. In this sense, telemedicine contributes to both safe and effective cancer care in the sense that follow-up was not interrupted, while vulnerable cancer patients were not exposed to the risk of contracting infections and concomitant complications. Examples of tele-oncology during the pandemic include the use of virtual tumor boards, remote monitoring of chemotherapy, and telepsychiatry for managing anxiety and depression. Relevant scholarship has also touched upon the facilitators and barriers to telemedicine adoption, which include access to technology and digital literacy, data protection and reimbursement policies [7].

Evidence from interviews involving healthcare workers and patients suggested that tele-oncology was superior to standard care in terms of convenience and reduced travel time and costs. Providers also reported improved patient engagement and satisfaction with telehealth, while survivors reported feeling safer and more comfortable at home. However, they also faced technical difficulties and limitations in physical assessments. Some providers also reported feeling less connected to their patients and were concerned about missing important nonverbal cues. On these grounds, it is expected that telemedicine will continue to play an important role in cancer care, particularly for survivors and for patients residing in remote communities [8].

### 1.3. Telehealth in Sarcoma Care

Tele-oncology for sarcomas often requires a multidisciplinary approach for proper management. Sarcomas are a diverse group of tumors that can arise from bone or soft tissue, and they represent less than 1% of all adult cancers and 15% of all pediatric cancers. The prognosis for sarcoma patients is highly dependent on the stage and histologic subtype of the tumor, with 5-year survival rates ranging from less than 10% for advanced metastatic disease to over 80% for early-stage disease [9,10]. One of the main concerns with tele-oncology for sarcomas is the potential for errors or delays in diagnosis, staging and treatment planning. Sarcomas can be difficult to diagnose due to their rarity and diverse histologic subtypes. Consequently, misdiagnosis or delay in diagnosis can have serious consequences for patient outcomes [11].

### 1.4. Aim

In a world gradually adopting tele-oncology, the need for ongoing evaluation and improvement of telehealth technologies and workflows is pivotal. The aim of this review is the assessment of current evidence regarding the utilization of telemedicine in diagnosis, treatment modalities, telerehabilitation and satisfaction among sarcoma patients and healthcare providers (HP).

## 2. Materials and Methods

The systematic review followed the recommendations of the Preferred Reporting Items for Systematic Reviews and Meta-Analyses (PRISMA). The protocol has not been registered. Data collection has been performed using the PubMed and Ovid MEDLINE databases with a defined search strategy from 2010 to date, according to the PRISMA recommendations [12]. Moreover, the websites and online repositories of the European Society of Oncology (ESMO), American Society of Clinical Oncology (ASCO), World Health Organization (WHO) and Cancer Research UK have been considered. All articles reported in scientific journals regarding the implications of telemedicine in sarcoma patients during the COVID-19 outbreak have been examined. After importing these articles into a reference management software (Rayyan), 1228 articles were gathered, of which 148 references were deemed eligible for inclusion in this qualitative synthesis and discussion (Figure 1). Subject headings included (“Telemedicine” or “Telehealth”) and (“Sarcoma”), and/or (“Cancer” or “Malignancies”), and/or (“Young adult cancer” or “Young adult malignancies”) and/or (“Ewing sarcoma”) and/or (“Osteosarcoma”), and/or (“Soft Tissue Sarcoma” or “STS”) and/or (“Gastrointestinal Stromal Tumors” or “GIST”), and/or (“COVID-19” or “SARS-CoV-2” or “Coronavirus disease 2019”) and/or (“Teleoncology” or “Tele-oncology”), and/or (“Telerehabilitation”). Abstracts from the ESMO and ASCO official websites have also been reviewed. Four independent investigators performed the final selection based on the relativity of the title and/or abstract, the publication date, the impact factor of the journal, as well as the language of publication. Non-English language studies were excluded, and all duplicates were removed. The search end date was 31 May 2023. An overview of the sampled articles including some basic information of them can be found in Appendix A.

## 3. Results

### 3.1. Qualitative and Quantitative Data Presentation

Our search yielded 1228 potentially eligible studies. After removing duplicates (*n* = 327) and screening titles and abstracts (*n* = 901), we deemed 176 studies eligible for full-text assessment. After removing those of them whose full text was not available in English or at all (*n* = 11), we examined the full text of 165 studies and eventually included 141. A PRISMA search flow diagram describing the search and selection process in detail can be found in Figure 1. A total of 22 out of these 141 articles were original research studies that included sarcoma patients. The characteristics of these 22 studies have been tabulated (Table 1).

#### 3.1.1. Telemedicine and Sarcoma Diagnosis

Delays in sarcomas diagnosis can be partially attributed to the negative impact of the pandemic on imaging case volumes. In particular, a retrospective trial demonstrated that the total imaging volume decreased by at least 28% over 7 weeks during the pandemic (weeks ten to sixteen) in comparison with 2019, including all healthcare service locations and imaging modalities. Statistically significant differences (*p*-value: 0.003) were observed in the mean weekly imaging case volumes after the pandemic onset, with a total volume decline of 24,383 [95% confidence interval (CI): 19,478–29,288] in 2020 versus 33,913 (95% CI: 33,429–34,396) in 2019. However, the impact on imaging volume varied by modality type. The greatest declines occurred in nuclear medicine (85%) and magnetic resonance imaging (MRI 74%), followed by ultrasound (US 64%), computed tomography (CT 46%) and X-ray (22%) [34]. These significant declines in MRI and US could detrimentally affect the early detection of sarcomas.

A retrospective study of 87 patients with bone sarcomas has more accurately evaluated the negative impact of the recent outbreak on the diagnostic process. Specifically, larger tumors (75 mm versus 55 mm, *p*-value: 0.025) and considerable biopsy delays (median: 12 days versus 6.5 days; *p*-value: 0.025) were documented in diagnoses before the pandemic compared with those during the outbreak, respectively. The difference in the median general delay between diagnoses before and after the onset of the pandemic was 3 months, but this finding did not reach statistical significance [31]. Another retrospective study including a larger sample of 372 patients with bone sarcomas and soft tissue sarcomas (STS) confirmed this delay until a definitive histologic diagnosis before and after the pandemic (90 versus 103 days, respectively, *p*-value: 0.024). However, no statistically significant differences were observed in terms of the stage at diagnosis, progression-free survival (PFS) and overall survival (OS) between the COVID-19 arm and the control arm [28].

#### 3.1.2. Telemedicine and Surgery

The COVID-19 pandemic has completely established a new reality in the treatment of cancer patients, creating new challenges spanning from a new diagnosis to follow-up. Most oncology societies in collaboration with the surgical ones responded with new guidelines focusing on the need for prioritization of patient triage, diagnosis, surgical and non-surgical therapy, as well as follow-up. Telehealth has been a huge aid in this endeavor of minimizing the effect of the pandemic on the treatment of cancer patients, with an estimated increase in its provision of 64.8% in 2020 compared to 2019 [6]. Telehealth in the form of video or telephone consultations, online questionnaires, multidisciplinary tumor boards (MDT) and medical apps plays a major role in reducing the exposure of both staff and cancer patients to a possible COVID-19 infection, thus protecting them, and also in conserving critical resources, such as intensive care units (ICU) beds and respirators in a time of need for medical systems worldwide [35]. These guidelines and suggestions aim at addressing the issue of promptly diagnosing new cancer cases, discussing these cases in multinational MDT, and establishing a therapy plan without delays and with a sense of involvement of both sides without compromising the satisfaction and needs of patients and healthcare practitioners. This was necessary to tackle the delays which around 70% of patients experienced [36].

In a large international cohort study about the postponement of surgery for 20.006 cancer patients across 15 countries, 10% did not undergo planned surgery for a follow-up of 30 weeks, and 4.5% were restaged [37]. At the beginning of the COVID-19 lockdown, there was an estimated 12-week delay for 37.7% of planned cancer operations across 190 countries according to experts [38]. A recent systematic review has reported that post-surgical mortality in patients with bone sarcomas and STS during the pandemic has increased with an odds ratio of 1.14. Delays in surgeries were associated with a worse surgical outcome, which was aggravated in the case of a COVID-19 infection [39]. All these data highlighted the need for an alternate response to COVID-19, and telehealth played a significant role.

A triage model was applied by Olshinka et al. in a quaternary-referral musculoskeletal oncology center in Quebec with the implementation of telemedicine in managing patients with low-grade malignancy or benign cases, which proved to be an efficient redirection of resources toward prioritizing malignant cases [13]. In some other cases, new strategies regarding sarcoma patients’ management have utilized non-surgical therapies in an attempt to better stratify the need for surgical intervention and respond to the new frame caused by the COVID-19 outbreak [40]. Despite the storm of COVID-19, which changed the approach in many cancers, high-malignancy sarcomas were given the utmost priority in the reviewed literature [41,42]. Another theme not highly addressed is the impact of COVID-19 on surgical training and burnout, which rose to threatening levels, and its impact on current and future delivered surgical care to cancer patients cannot be efficiently judged [43].

Despite the plethora of guidelines, recommendations and suggested triage models, the international approach in prioritizing sarcoma surgery was not adequate, as depicted by the worldwide survey among 152 musculoskeletal oncology surgeons who reported a major delay in 20% of their cases for life-threatening sarcomas [44]. The complexity of some cases, the limitation of available resources and the imposed regulations by governments created a tough background, testing the commitment of oncologic surgeons to ensure a prompt response to sarcoma cases [45]. This highlighted the need for a rearrangement of medical systems worldwide, and telehealth seems to be a major player in this newly formed vision. A content summary of guidelines and practical recommendations for sarcoma patients during the COVID-19 outbreak can be found in Section 3.2.

#### 3.1.3. Tele-Oncology for Sarcoma Patients

Tele-oncology is defined as the employment of telemedicine technology to deliver cancer care services to patients in remote locations, which includes the areas of diagnosis, treatment, supportive care and even prevention of cancer [46]. It involves a plethora of virtual means, such as web conferencing via Zoom and other mobile applications, as well as remote chemotherapy supervision models, that enable physicians to provide high-quality clinical oncology services from a distance [3]. This new entity has already been adopted by many healthcare centers contributing to the revitalization of traditional medical services [47]. A striking example is that face-to-face consultations have been substituted by videoconferencing with the intent to facilitate rural populations. Tele-oncology entails a home-based environment that can potentially guarantee a higher level of compliance concerning lifestyle changes, treatment strategy, symptom control and overall management [48].

However, tele-oncology had not been broadly adopted, with the exemption of patients in remote areas, until the COVID-19 pandemic onset [49]. Thereafter, its increasing implementation has led to fewer outpatient visits and, thus, a decreased risk of exposure to the SARS-CoV-2 virus [50]. A 7.7-fold increase in virtual consultations had been observed during the pandemic, and 25.1% of specialists were concerned that this increase would be associated with a negative impact on their patients’ survival [36]. A survey that was conducted in oncological departments in 12 of the most affected countries in Europe and the USA reported that telemedicine had been implemented in 76.2% of centers. Video-conferencing for MDT was carried out in 81% of centers [51]. Another cross-sectional study regarding radiation oncology practice showed that the frequency of video consultations was considered by 67% of radiotherapists and 30.6% of patients to be inadequate [52]. A survey by the American Society for Radiation Oncology has demonstrated that the majority of the radiation oncology units (89%) provided telemedicine consultations to their patients during the pandemic [53]. According to the aforementioned ESMO recommendations, “low priority” patients are highly recommended to consider telemedicine services [54,55,56]. Remarkably, telemedicine includes virtual care and virtual prescription of treatments, as well as the subsequent administration of medication when it is feasible [57]. The implementation of an agile virtual care program at one of the largest cancer centers in Canada has yielded promising results. In particular, high outpatient caseloads were preserved without affecting the quality of care, as has been confirmed by the high satisfaction rates of both HP (72%) and patients (82%). This initiative was also associated with significant displacement-related cost savings for patients [58]. High satisfaction rates (83%) were documented in another survey including patients with various cancer types. Of note, participants older than 50 years old were less likely to be satisfied with virtual oncology appointments (adjusted odds ratio, 0.22; 95% confidence interval, 0.66 to 1.07) [59]. Moreover, telemedicine integrative oncology consultations have been shown to contribute to fewer symptom management concerns in comparison with conventional appointments [60].

As far as sarcoma patients are concerned, the multidisciplinary nature of sarcoma care requires close coordination between medical and surgical oncologists, radiation oncologists, and pathologists, which can be challenging to achieve in telehealth settings [61]. An online survey assessed the satisfaction rate of physicians from a tertiary sarcoma center in the UK concerning their experience with virtual MDT. Overall, 83.3% of participants considered that virtual MDT did not affect the decision-making process compared with in-person consultations, and 72.8% of specialists were satisfied with this new opportunity. The majority of them (55.6%) answered that they would prefer to attend only virtual MDT in the future, whereas approximately 92% recognized that this novel medical model could serve as the inception of global virtual MDT involving physicians from abroad in the near future [25].

Additional factors that complicate sarcoma patients’ management include their high rates of severe COVID-19 infection and the consequences of a delayed diagnosis. High rates of ICU admissions and mechanical ventilation (11% and 6%, respectively) were observed in this population, whereas bone sarcoma patients with COVID-19 infection demonstrated the highest mortality rate among sarcoma patients, according to a retrospective trial [32]. Another retrospective study of 223 patients with STS revealed that delays ≥ 1 month in treatment, due to the pandemic, caused a higher distant metastasis-free survival, but it was not associated with local recurrence-free survival or disease-specific survival [33].

However, telemedicine seems to be a viable and effective tool for both HP and sarcoma patients, as it has been underlined in several satisfaction surveys tailored for this subpopulation. An online survey showed comparable patients’ satisfaction rates between virtual appointments (94%) and in-person appointments (95%). On the other hand, 88% of patients were vehemently opposed to being informed about “bad news” through telemedicine [11]. The percentage of patients who would prefer face-to-face consultations for the “bad news” announcement was significantly lower (48%) in another survey that included patients with rare cancers. A high satisfaction rate (90%) was observed in this survey, as well. Consequently, flexible and mixed care delivery seems to be the ideal approach toward these patients [15].

As far as anticancer therapy for sarcoma patients is concerned, targeted treatment has revolutionized the treatment landscape in gastrointestinal stromal tumors (GIST) and STS. These agents are administered per os, and thus, virtual prescription per se can lessen the number of hospital visits and the risk of exposure to SARS-CoV-2 [53]. Typical examples of approved targeted agents include (a) imatinib for GIST, dermatofibrosarcoma protuberans and tenosynovial giant cell tumor, (b) regorafenib for GIST and osteosarcomas, (c) larotrectinib for neurotrophic tropomyosin receptor kinase (NTRK) fusion-driven sarcomas, (d) sunitinib, (e) cabozantinib, (f) ripretinib and (g) avapritinib for GIST, as well as (h) pazopanib for non-adipocytic STS and (i) sirolimus for perivascular epithelioid cell tumors [62,63,64,65,66,67,68,69,70,71,72,73]. Moreover, hormone therapy with a per os aromatase inhibitor is indicated for endometrial stromal sarcoma [74]. Concerning chemotherapy in the era of COVID-19, it is worth mentioning that a retrospective study of pediatric Ewing sarcoma patients evaluated the feasibility and safety of the outpatient administration of ifosfamide etoposide. It was reported that 74% of cycles were administered on time, and no event of hemorrhagic cystitis occurred (monitoring via urine dipstick) [29].

Finally, the implementation of telehealth was the cornerstone of the ESMO multidisciplinary expert consensus regarding the management of cancer patients during the COVID-19 outbreak. Besides the aforementioned patients’ prioritization, the significance of telehealth adoption in daily clinical practice was emphatically highlighted. In particular, telemedicine and digital health in oncology were recommended to be implemented for virtual appointments and primary care triage, as well as for important interventions, such as virtual counseling, virtual prescriptions, management of long-term therapy and follow-up. Last but not least, it is underlined that telemedicine effectively promotes wellness interventions including health education, health risk assessment, nutritional monitoring, physical exercise, mental fitness and medication adherence, thus achieving a higher quality of life (QoL) [54].

#### 3.1.4. Telerehabilitation for Sarcoma Patients

With the term telerehabilitation, we refer to the use of telehealth media in delivering a multitude of rehabilitation services to patients from a distance. The optimization of telerehabilitation services through Internet-based models increases access and allows the rehabilitation therapist or clinician to tailor treatment in a personalized manner, resulting in enhanced adherence and outcome [75]. Rehabilitation services for cancer survivors include a great spectrum of simple consultations, physical therapy, symptom management, psychotherapy, as well as occupational and speech therapy, aiming at lessening the burden of cancer while maintaining a good health-related QoL [76]. Before the pandemic, telehealth was employed in the delivery of rehabilitation services mostly in the form of providing emotional support, symptom management and physical therapy. The COVID-19 pandemic was associated with an increasing frequency of telerehabilitation services, highlighting the trend of providing an individualized care system for cancer survivors, bypassing face-to-face interventions, and creating an exemplary individualized asynchronous care system [77].

Exercise is universally recommended for cancer patients as it reduces the risk of all-cause mortality in many cancers, improves muscle strength via antagonizing cancer-bound cachexia and muscle wasting and improves almost all QoL indexes [78]. Internet-based platforms are used on providing home-based exercise interventions, even yoga sessions to cancer patients. Messages and videoconferences which are shared on the platform aid research staff in monitoring patients’ progress and adherence, as well as in providing individualized feedback [79]. Randomized control trials in patients with breast cancer following this model showed adherence and improved satisfaction scores compared to face-to-face consultations [80,81]. A review incorporating 3600 patients from distinct populations validated the role of telehealth interventions as a means to increase the effectiveness of rehabilitation [82]. Another pioneering telehealth step is the use of virtual reality technology in rehabilitation through the implementation of physical therapy services through augmented reality, which showed many positive results [83].

The psychological trauma of the COVID-19 pandemic and the imposed lockdown has been associated with severe complications for cancer patients, and it is often underestimated. The diagnosis of cancer, especially of sarcoma, which constitutes a complex disease, is linked with unpleasant emotions that have been exacerbated in the time of COVID-19. As a result, almost 25% of sarcoma patients reported psychological distress and a level of personal unsafety and fear that often interfered with their daily functioning [11,84]. Telehealth may address this psychological stress through interventions that include online self-help training based on cognitive behavioral therapy, an online symptom self-management curriculum that teaches coping skills, and even mobile games. Therefore, it has been proven effective in reducing depression, anxiety and improving cognitive function, as well as gaining optimism [85]. It has been reported by Samson-Daly et al. and Melon et al. that telehealth is useful in young patients with cancer, as this population experiences more severe physiological distress and has fewer coping mechanisms [16,86,87]. However, regarding the implementation of telehealth in psychological support, there are certain problems needed to be addressed, pertaining to the unique characteristics of malignancies under therapy [76]. For example, sarcomas as unique entities require a more detailed approach that is tailored to the patient’s individualized needs [28].

Cancer patients in all phases of their malignancy experience symptoms either relating to the primary disease or those that are secondary to the treatment. Through consultations, applications or other telehealth services has assisted cancer patients in handling these symptoms. A review and meta-analysis, which included nine studies, showed that patients receiving telerehabilitation services had a QoL and control over their symptoms relative to those who received face-to-face rehabilitation services [88]. Basch et al. reported that electronic symptom monitoring through Internet-based applications and the corresponding feedback resulted in an improvement in the QoL, as the patients received individualized directions on how to approach their symptoms [89]. Many observational and RCT studies implemented augmented reality technology to evaluate the interventions for pain, fatigue and cognitive impairment, which show favorable outcomes [90,91,92].

Concluding, telehealth has pioneered rehabilitation by enriching the spectrum of offered services and supporting the patients’ needs in this new landscape. What seems to be lacking is the confrontation with palliative care, especially in patients over 60 with limited access to telehealth and the individualization of offered services to some rare cancers such as sarcomas [93].

#### 3.1.5. Telemedicine and Mental Health during COVID-19

The COVID-19 pandemic created an environment with many risk factors pertaining to mental health problems. The unpredictability of the situation, the uncertainty along with measures leading to social isolation, loss of income, loneliness, inactivity and limited access to basic necessities led to a soar in mental health problems [94]. Cancer patients represent a particularly vulnerable population, which was severely affected by the pandemic. As previously mentioned, around 70% of cancer patients experienced a delay in their planned procedures, which combined with the feeling of isolation from their health practitioner and the economic downturn with financial insecurity and unemployment portend poor mental health outcomes [36]. In the literature, there is a focus on the mental burden of cancer patients as a vulnerable population and rarely categorizes them into specific cancer types.

Younger et al. have investigated the psychological impact of the pandemic on sarcoma patients. In this survey of 350 patients with sarcoma, an increase in loneliness, feelings of worry and insomnia were reported, especially among participants in palliative care or without a clear treatment intent. A total of 259 out of 350 patients (74%) indicated a preference for telehealth appointments in the future compared with 78 patients (22%) that would prefer only face-to-face appointments. The treatment intent and the resilient coping level were significantly related to the preference for only in-person appointments (*p*-value: 0.047 and 0.024, respectively). In particular, patients with unknown treatment intent or a low resilient coping score were most likely to prefer face-to-face contact. A decrease in all aspects of all health-related QoL (HRQoL) concerning physical, emotional and social functioning compared to pre-pandemic was also documented [14].

Telemedicine offered a plethora of tools in assisting cancer patients and, by extension, their support networks, facing challenges regarding their mental health. From applications that track patients’ mental health to psychoeducation as well as organizing videoconferences with psychiatrists and psychologists, the technological advances offered many choices that were recommended by cancer societies worldwide. Many hospitals at the epicenter of the COVID-19 outbreak set up platforms for oncologic patients’ psychiatric assessment with group sessions, education on behavioral therapy and regular follow-up. These initiatives were appreciated by both sides (healthcare professionals and patients), with significant satisfaction reported from patients who could convey their feelings of stress and uneasiness from the comfort of their homes and with support from their family members [95].

Singleton et al., in a meta-analysis of eHealth interventions for patients with breast cancer, reported improved patient-reported outcomes, a better QoL and lower emotional distress with an improved response to tele-interventions which variated from group sessions with a psychologist to behavioral therapy. However, there is a lack of evidence specifically focused on cancer patients after the pandemic, whereas the current ones have only shown positive short-term outcomes. Healthcare professionals have a strategic role in promoting tele-psychology interventions, addressing the current issues of privacy and ensuring that this therapeutic relationship remains respected [96].

#### 3.1.6. Compliance and Satisfaction with Telemedicine

Even prior to the COVID-19 pandemic, there were several studies and systematic reviews examining the implementation of telemedicine and the satisfaction of both patients and HP [97,98]. The use of web platforms, video conferences and applications seemed to balance the lack of face-to-face medical care by reducing the travel burden of cancer patients while enabling timely clinical interventions and the treatment plan to be carried out. Initially, there were many challenges and concerns for the use of telehealth among cancer patients and healthcare workers, but with time and correct implementation of the system from both sides, the goal of a high patient care continuum seems to be achievable [53].

In particular, the transformation of traditional MDT to virtual MDT was met with doubt. However, Rajasekaran et al. reported that the use of virtual MDT in the United Kingdom had positive results, with 75% of participants being satisfied, and thought this to be a precursor of global virtual MDT [25]. One tactic regarding global virtual MDT is that organized by the ESMO, with specialized individuals representing all cancer types. Generally, healthcare providers endorsed tele-oncology and showed high rates of satisfaction in establishing HP–patient relationships, allocating resources and coordinating with other HP [30,99]. However, there was some concern about achieving mutual trust in an HP–patient relationship, especially in newer patients without in-person interaction. This was somehow mitigated through video consultations, which most HP regard as more beneficial for maintaining a high level of care [100].

As far as patients are concerned, there are plenty of studies utilizing different forms of telehealth with intent to evaluate their satisfaction during the follow-up and even their physical examination competency. Reducing transport, especially for sarcoma patients who need to travel long distances to highly competent medical centers, being informed in the comfort of their own home in a convenient and supportive environment and simultaneously being protected from COVID-19 have been documented to be efficient and have generally been praised [35,53,88,97]. Telehealth has been proven to be an important tool in supporting patients with palliative care, establishing a comfortable environment and enhancing the feeling of security and safety [93]. However, patients were opposed to being told bad news or examination results through telemedicine, were unsatisfied with the lack of physical examination, which even led to adverse effects and feelings of physiological distress when there was a dispute between HP [11,15,23,101]. Telehealth experiences were also worse for patients from a low socioeconomic zone or a history of anxiety or depression [102].

Furthermore, a cross-sectional survey of 350 sarcoma patients evaluated the psychosocial impact of care modifications during the initial phase of the COVID-19 pandemic. Overall, 72% of participants considered that the quality of care had not been affected, but 87% and 41% of patients reported a negative impact on their social life and emotional well-being, respectively. Interestingly, 34% and 31% of patients postponed their follow-up appointments and scans, respectively, for more than three months. In addition, 30% and 41% of these patients were disappointed with the decision to postpone their appointments and scans, respectively. Postponements regarding follow-up were mainly observed in the elderly (43%), whereas therapy postponements were significantly associated with treatment intent (*p*-value: 0.0001) and most commonly occurred in patients who received palliative treatment. COVID-19 infection-related worry was significantly associated with sarcoma-related worry, which led to deteriorated cognitive and emotional functioning. Higher scores of worry, insomnia and overall health QoL were also documented in patients who were not aware of their treatment intent. Providing psychological support to these patients is of paramount importance during the COVID-19 outbreak and beyond [14].

In general, telemedicine enriched the medical arsenal in reversing the damage caused by the COVID-19 outbreak in sarcoma patients. Both sides reported that the correct implementation of telehealth can result in a patient care continuum proportional to face-to-face health services. The hurdles needing to be overcome are limitations with technology, reduced confidence in the doctor and the inability to be physically examined. By utilizing the advantages of telehealth, with individualized care and combining telehealth with physical visits, not only can we create a nourishing environment for cancer patients, but also a health system of which HP and patients are praising advocates.

### 3.2. Telemedicine Guidelines, Practice and Policy

The complex and multiform nature of sarcomas designates the modernization of available guidelines as an absolute necessity, as they require a multidisciplinary approach from many planes of available health services. After the outbreak of the COVID-19 pandemic, there was an extensive release of new guidelines and recommendations among many oncologic societies (e.g., ASCO, ESMO), which were released after a consensus among international experts pertaining to all steps of cancer therapy, aiming at harnessing the positives of our technological society, thus establishing a continuum in patient care. Most recommendations categorize patients into high, high/medium, medium and low priority according to their primary disease and their metastatic state [54,103,104,105,106,107]. Regarding surgical therapy, patients with Ewing sarcoma, osteosarcoma, rhabdomyosarcoma and high-risk STS and GIST should be treated according to MDTs discussion, and their resection is not to be postponed [108,109]. Moreover, biopsies for undetermined mass or osteolysis with a high suspicion of malignancy should not be deterred. Fixation of pathological fractures and spinal cord decompression remained established as a non-deferrable operation [110]. For patients with benign biopsies, primary localized low-risk sarcomas, mesenchymal tumors of intermediate malignancy or GIST tumors which are stable on maintenance tyrosine kinase inhibitors (TKIs), therapy could be deferred for 3–4 months or be distributed to local services [105].

Regarding radiation oncology, therapy should not be postponed for sarcomas with a high recurrence risk. RT is considered a high priority in the therapeutic approach toward Ewing patients even during the pandemic, according to various guidelines [106,111]. RT was also deemed as an utmost priority for any symptomatic metastatic lesion that could be alleviated by means of RT. For low-grade STS and asymptomatic primary tumors, there was a consensus for postponement of RT according to available resources but not over 3 months [105,106]. In general, neoadjuvant RT for STS is associated with wound complications after RT. However, some patients with large border-line resectable sarcomas may reap the benefits from preoperative RT, including the lower risk of exposure to SARS-CoV-2 and the decreased overall time treatment, as well as the lower risk of tumor cell seeding during operation [112].

Neoadjuvant chemotherapy for extraskeletal Ewing sarcoma, high-risk STS and paediatric-type rhabdomyosarcoma or osteosarcoma should be further advanced according to MDTs. Imatinib should be given to locally advanced GIST with sensitive genotype or >40% recurrence risk without delay. As far as the locoregional GIST is concerned, adjuvant treatment with imatinib is the standard of care for imatinib-sensitive GIST patients with a high risk of relapse based on their mutational status. Imatinib, sunitinib, regorafenib, repretinib, as well as larotrectinib or entrectinib are included in the therapeutic armamentarium regarding metastatic GIST [108]. The virtual prescription of all these orally administered agents is associated with multiple benefits for GIST patients in terms of decreased exposure to infectious agents and QoL [53]. Perioperative chemotherapy for high-grade bone sarcoma, chondrosarcoma and beyond the third line in Ewing sarcoma was considered low priority and could be transitioned for a delay, as well as the use of imatinib in GIST with <40% recurrence risk [106]. Furthermore, there was a consensus of continuing RT on an outpatient basis as a bridge therapy to postpone therapy in an effort to allocate available resources, even when the treatment is not standard, provided it is on a no-harm basis. The use of neoadjuvant imatinib in GIST-sensitive tumors can also be considered a bridge therapy even without a formal indication [105].

Of note, the ESMO multidisciplinary expert consensus agreed on some clinical statements regarding the management of cancer patients during the pandemic. The first statement regards telehealth, which is considered a useful tool for facilitating the management of cancer patients. It is also suggested that clinicians expand the duration of granulocyte colony-stimulating factors (G-CSF) in order to lower the risk of febrile neutropenia in high-risk groups. Thromboprophylaxis for patients with cancer and COVID-19 is highly recommended [54]. According to the updated ESMO practice guidelines concerning bone sarcomas, the recommended frequency of FU visits is lower for low-grade bone sarcomas compared with high-grade tumors [109]. The ASCO has also published a guide for cancer care delivery during the COVID-19 outbreak. This report focuses on telemedicine implications with regard to FU, per os anticancer treatment adherence, genetic counseling, patient education, survivorship, palliative and supportive care, symptom monitoring, triage and quick patient assessment. According to the ASCO, home infusion of chemotherapy is not generally recommended. However, home infusion of antiemetics and hydration could be a considerable option [104].

Moreover, the International Cardio-Oncology Society (ICOS) has reported specific guidelines regarding cardio-oncology patients in the era of COVID-19. Telemedicine (e.g., virtual consultations) and limited cardiovascular imaging are recommended for routine cardiovascular surveillance of low-risk patients. Given that the cornerstone of sarcoma patients’ chemotherapy consists of cardiotoxic anthracyclines, the ICOS statement provides substantial insights. As far as patients who have received RT are concerned, clinicians should be aware of their higher risk for a complicated course of COVID-19 due to their reduced cardiopulmonary reserve. GIST patients who are treated with TKIs, such as regorafenib, should undergo routine home blood pressure monitoring. Cardio-oncology patients with impaired Karnofsky performance status and COVID-19 infection should receive anti-thrombotic prophylaxis [113].

Last but not least, online practice guidelines have been developed by the Society for Integrative Oncology (SIO) with a view to ensuring continuity of care beyond the COVID-19 pandemic. These online practice recommendations aim at addressing challenging issues regarding online consultation, symptom management and treatment for oncological patients. HP should identify and handle potential skepticism toward the effectiveness of a virtual oncology treatment program [114]. A detailed overview of all the aforementioned recommendations for sarcoma patients during the pandemic is illustrated in Table 2.

## 4. Discussion

### 4.1. Benefits of Telemedicine in Cancer Care

Telemedicine is a promising modality that has been involved in the oncological clinical practice for the last twenty years [115]. The embracing of tele-oncology by hospitals and medical centers in terms of cancer care has been tested in a wide umbrella of models that complement or subsidize physical consultations. This is especially important for outreach services for patients that reside in remote areas and face difficulties in reaching a cancer center [49]. By incorporating telemedicine technologies in daily clinical practice, equality is preserved among patients regarding access to appropriate oncologic management. The new era in oncology demands shifting to a patient-centered care which implies implementing advanced methods in order to blunt inequalities among rural populations, as well as those among people in difficult financial conditions [116]. Out-of-pocket expenses for travel to centrally located cancer institutes, as well as lodging and missed work, can only decrease the overall quality of cancer care. This could be avoided with the effective incorporation of tele-oncology in standard-of-care practice [117,118].

Another important issue that can be addressed with the wise use of technology regards screening and early diagnosis. The COVID-19 pandemic turned the spotlight on the importance of minimizing the exposure risk of potential patients to the SARS-CoV-2 virus and hospital-acquired infections in general. The implementation of telemedicine has succeeded in reassuring a significant part of the appropriate cancer management and follow-up without the added risk of exposure [53].

This applies even more to patients who are immunocompromised because of cancer. Unnecessary outpatient visits can be replaced with video conference sessions or telephone consultations [119]. The formation of appropriate tailor-made therapy plans for patients, as well as timely clinical intervention and management, is of utmost importance. Not only does telehealth offer the opportunity for synchronous interactions between patient and doctor, but it also facilitates videoconferencing bringing together a multidisciplinary clinical team. Thus, it is feasible for a patient to have access to clinicians of expertise without the potential subsequent delay to one’s treatment plan. Remote supervision of oral chemotherapeutic treatments and radiation oncology can also be applied via telemedicine [48].

The application of telehealth can also be in favor of reducing perioperative and postoperative stays in the hospital setting, as well as avoiding surgery delays when it comes to management. Telehealth has the potential to reduce the infection risk, as well as the risk of chemotherapy-induced myelosuppression in high-risk patients [115]. Remote monitoring of adverse effects, remote symptom management and easier transmission of clinical information such as imaging data, laboratory data and histopathological results can only optimize patients’ cancer care without necessitating hospitalization [120].

The aforementioned initiative is beneficial to patients with pre-existing mental health issues, as well as to immunocompromised patients in terms of infection control and protection. A prolonged hospitalization period implicates the arousal of feelings of insecurity, fear and solitude. The psychological challenges that cancer patients need to confront can be alleviated with the implementation of telemedicine [121]. Internet-based provision of emotional and psychological support, as well as the relief and reassurance that a healthcare provider can easily be accessed at any time from the comfort of one’s home, are factors that contribute vitally to a patient’s rehabilitation plan [122].

Telerehabilitation comprises distant monitoring of potential adverse effects of treatment by clinicians, symptom follow-up and respective management, pain relief, personalized speech or physical therapy programs with the appropriate guidance of an expert, as well as psychosociological aid [123,124]. Communication technologies such as mobile phone applications or online video session platforms can facilitate reasonable medical care access to cancer patients from the comfort of their living environments [3]. Telehealth is capable, in this setting, of reassuring adequate and in-time contact with patients in the follow-up phase, meeting the challenge of limited personnel and few financial resources. Moreover, it enhances the process of recovery and attains, therefore, better QoL leading to improved health outcomes [123]. Finally, telemedicine can be beneficial to patients in long-term follow-up in financial terms, given that it can alleviate travel and lodging expenses and reduce missed work days [115].

### 4.2. Challenges of Telemedicine

Telemedicine has a major—and, to an extent, documented—potential to improve access to care for patients with sarcoma, especially those in remote areas or with limited mobility and caregiving support. However, there are a number of challenges associated with telemedicine regarding diagnosis, treatment and monitoring.

Outsourcing the diagnosis and investigation process to largely remote modalities poses a risk of gathering insufficient information. Addressing patients’ complaints before the diagnosis of a sarcoma through online modalities can lead to misdiagnosis or a delay in appropriate investigation, in case symptoms or family history are not correctly iterated during the interview. Risks apply to patients with an established diagnosis that receive treatment or require follow-up. Many aspects of patients’ physical, mental and social history can be missed during online consultations or poorly transferred to oncologists or members of oncological councils deciding the course of treatment. This could lead to delayed or inefficient management [125]. The lack of personal connection and inability to provide a physical examination in telemedicine settings prevents physicians not only from achieving objective findings, such as lymphadenopathy, suggestive of metastatic disease, but also from gaining the trust of the patient and understanding their environment. An incomplete patient–physician alliance can only be considered as an ill-omen at the beginning of an undoubtedly arduous therapeutic process [126]. Both of them are compounded by the lack of knowledge in the form of peer-reviewed publications or practice guidelines. Building a scholarship out of good practices, outcomes and even mistakes will take years. During this time, telecare for sarcoma can only be practiced with a great deal of uncertainty [11].

The challenges facing telecare for sarcoma need to be underscored from a population health point of view as well. Telemedicine can be undermined by diverse inequalities. Digital literacy gaps between older and younger individuals, and technology and Internet access gaps between high- and middle-and-low-income countries condemn the less tech-savvy and the non-privileged populations to suboptimal remote care [127]. This situation can be further exacerbated by considering a similar digital divide among physicians and healthcare providers. Therefore, telecare for sarcoma can achieve its potential providing that both patients and physicians have sufficient command of and access to digital services [128,129].

However, sarcoma patients often receive complex treatment regimens that require admission and hospitalization. The pandemic caused significant interruptions to these patients’ therapeutic plans, but telehealth cannot effectively address these emerging issues. A striking example is that many patients with complex radiation and chemotherapy schedules were required to either modify their regimen or delay their treatment. Another considerable limitation of this group of patients was the difficulty in obtaining laboratory or radiologic imaging studies [130].

Finally, yet importantly, regulatory disparities need to be considered. National legislations regarding patients’ privacy differ between healthcare systems, countries and continents. It is reasonable to assume that time-consuming approval processes can delay critical clinical decisions and demotivate patients and physicians. Conversely, approaches allowing a virtually unlimited flow of information pose a threat to patients, in the event that their health records and personal struggle during the diagnosis and treatment become accessible to third parties such as future insurers or employers. The latter concerns not only the patients but all those who share a household with them—caregivers and family members can see the privacy of their coping process being diminished with uncontrollable consequences in the future. Equally, physicians, aware of potential retribution for any words they utter, are more likely to hesitate to engage in conversations delving into the patients’ and caregivers’ sentimental and practical struggles [131,132].

Despite these challenges, telemedicine has the potential to improve access to care for patients with sarcoma. Studies are starting to test how to transition telehealth from a temporary solution during the pandemic to a permanent part of cancer care. To overcome the challenges, healthcare providers and patients need to be educated on the use of telemedicine, and the technology needs to be improved to ensure that it is user-friendly and accessible to all patients (Figure 2) [133].

### 4.3. Opportunities and Future Perspectives

On 5 May 2023, the Director General of the WHO declared the end of the COVID-19 pandemic as a global health emergency [134]. By that time, a significant amount of healthcare delivery had returned to its initial physical settings. Nevertheless, the obtained experience and infrastructure for telecare in cancer, and particularly sarcoma, constitutes an asset that can be further availed in the post-COVID-19 era.

The opportunities of telemedicine include (a) improved access to specialists for people residing in remote areas; (b) remote and potentially more regular consultations and monitoring, given the decrease in transportation time and costs; and (c) pragmatic and/or home-based clinical trials, where patients can be monitored in their daily living settings in accordance to the principles outlined through the concept of living labs [11,135]. The decrease in waiting time and transportation costs can also be fueled by higher compensation awarded to volunteer subjects in phase 1 clinical trials or researchers, who tend to be suboptimally compensated in several low- and middle-income countries [136]. The latter can be considered an opportunity to dismantle the gender pay gap in oncology [137].

The aforementioned advantages apply in the context of pandemic preparedness and broader health crises. In the event of emerging epidemics, telecare, a concept that is actively researched by health bodies such as the WHO or regulatory authorities such as the European Commission, could provide for the continuity of care for patients with sarcoma. As a matter of fact, both telemedicine and non-communicable diseases (NCDs) such as cancer consist of the cornerstones of the EU4Health Programme, the European strategy for containing the cross-border transmission of pathogens [138].

Broader crises with dire implications in healthcare, such as war, have shown that telemedicine holds great potential in cancer care for internally displaced populations and refugees. The effectiveness of telemedicine intervention in this case is certainly subject to limiting factors such as the availability of telecommunications and injuries sustained within the conflict either on the patients’ or the healthcare providers’ side [139]. These limitations apply to regions neighboring conflict sites—the energy crisis following the Russia–Ukraine war in 2022 serves as an example of a potential stalemate, where energy shortages, in the form of either blackouts or unaffordable energy costs, could deplete remote populations from remote consultation and care [140]. A response to such a situation would necessitate a liaison between tele-oncology and renewable forms of energy. Such a step would not only increase the sustainability of remote sarcoma care, but it could also reduce the carbon footprint of healthcare, an industry with a steadily high contribution to the climate crisis [141].

Overall, telemedicine has the potential to improve the diagnosis and care of sarcoma patients by improving access to care, reducing treatment delays, and improving patient outcomes. However, more research and guidance in the form of practice guidelines is needed to fully understand the potential benefits and limitations of telemedicine and champion good remote practices in sarcoma care.

## 5. Conclusions

The COVID-19 pandemic has been associated with a detrimental impact on health services. On the other hand, every cloud has a silver lining, and thus, the COVID-19 outbreak has accelerated the adoption of telehealth and has taken the digital transformation of medicine to a whole new level. The development of telehealth has the potential to rejuvenate daily practice given that its implications will affect a wide spectrum of cancer patients either under active treatment or surveillance. This new reality can apply even to malignancies of surgical interest, such as sarcomas. The maintenance and evolution of telemedicine can be proven beneficial to this special patient population in terms of QoL beyond the pandemic, as well. However, further investigation is required to assess sarcoma patients’ experience of the emerging electronic platforms, as well as to establish their non-inferiority compared with conventional clinical practice.

## Figures and Tables

**Figure 1 cancers-15-03700-f001:**
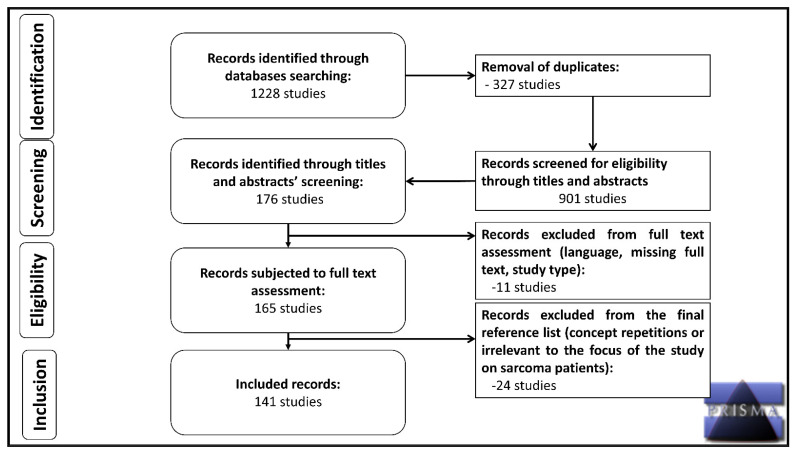
Flow diagram based on the PRISMA checklist.

**Figure 2 cancers-15-03700-f002:**
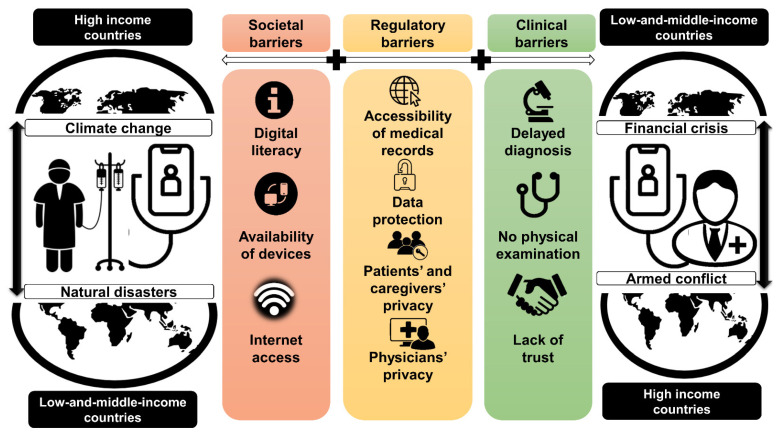
Challenges regarding telemedicine. Societal, regulatory and clinical factors pose barriers to remote sarcoma care being compounded by structural inequalities across high- and low-and-middle-income countries and emerging political, financial and environmental crises.

**Table 1 cancers-15-03700-t001:** Overview of the characteristics of sarcoma-related studies during the pandemic.

First Author (Study)	Date	Study Type	Cancer Type	Sample	Type of Telemedical Intervention	Phase of Cancer Care	Categorization
Olshinka et al. [13]	2020	Retrospective observational study	Sarcoma	155 FUs, 96 new referrals	Email-fax-telephone	New referrals, FU	Surgery, TO
Younger et al. [14]	2020	Cross-sectional questionnaire-based survey	Sarcoma	350 patients	N/A	Under treatment	Mental health, Satisfaction
Smrke et al. [15]	2020	Observational, questionnaire-based survey	Sarcoma	316 patients (379 appointments), 18 providers and 108 patients completed the survey	Telephone	New referrals, under treatment, FU	Satisfaction, TO
Lidington et al. [16]	2020	Retrospective, observational study	Sarcoma	350 patients	N/A	Under treatment	TR, TO
Buonaguro et al. [17]	2020	Retrospective, observational study	Various types including sarcoma	N/A	Telephone, e-mail, a home healthcare platform, intensive healthcare platform	Under treatment, FU	TH in oncology and COVID-19, TO
Martinez et al. [18] American Society for Radiation Oncology	2020	Questionnaire-based survey	Various types including sarcoma	115 responses	N/A	Under treatment, FU	Teleradiology (TO)
De Joode et al. [19]	2020	Survey-based analysis	Various types including sarcoma	5302 patients	Telephone, video	Under treatment, FU	Satisfaction, mental health
Košir et al. [20]	2020	Cross-sectional study	Various types (AΥA)	177 individuals	N/A	Under treatment	TO, Mental health
McCabe et al. [11]	2021	Comparative questionnaire-based observational survey,	Sarcoma	74 patients, 26 providers	Telephone or video consultation	Under treatment, FU	TR, TO
Folsom et al. [21]	2021	Case series	Various types including sarcoma	2 patients	videocall (zoom)	Under treatment, FU	Satisfaction, mental health
Hassan et al. [22]	2021	Case report- observational	Leiomyosarcoma	1 patient	N/A	Diagnosis	Diagnosis, TO
Lawrenz et al. [23]	2021	Cohort-telephone survey	Sarcoma	64 patients	Telephone	FU	Satisfaction, TO
Yeshayahet al. [24]	2021	Case series	Ewing Sarcoma	4 patients	Telephone consultation before ER	New referrals	Diagnosis
Rajasekaran et al. [25]	2021	Questionnaire-based survey	Sarcoma	36/39 responses	Videoconferencing platforms	MDT	Satisfaction, TO
Natesan et al. [26]	2021	Questionnaire-based survey	Various types including sarcoma	18 radiation oncology providers	Telephone or video	Under treatment	Satisfaction, teleradiology (TO)
Global Health Research Group on Children’s Non-Communicable Diseases Collaborative [27]	2022	Retrospective observational cohort study	Various pediatric tumors including sarcoma	1660 patients (91 hospitals and cancer centers in 39 countries)	N/A	New referrals, under treatment	TO
Onesti et al. [28]	2022	Retrospective study	STS and bone sarcoma	372 patients	N/A	Diagnosis	TR, TO
Sarfraz et al. [29]	2022	Retrospective study	Ewing sarcoma	20 patients	N/A	Under treatment	TO
M van Erkel et al. [30]	2022	Qualitative study using semi-structured interviews and reflexive thematic analysis	Various types including sarcoma	82 patients and 58 providers	Telephone consultation before ER	FU	Satisfaction, TO
Kotrych et al. [31]	2022	Retrospective study	Bone sarcoma	87 patients	N/A	New referrals	Diagnosis
Wagner et al. [32]	2022	CCC19-Registry Based Retrospective Cohort Study	Sarcoma	281 patients	N/A	N/A	TO, sarcoma and COVID-19
Hong Ryu et al. [33]	2023	Retrospective trial	Sarcoma	5927 (223 eligible patients)	N/A	New referrals	TO

Overview of the characteristics of the studies that included sarcoma patients during the COVID-19 outbreak, in chronological order. AΥA: adolescent and young adult malignancies; CCC19-registry: COVID-19 and Cancer Consortium Registry; ER: emergency room; FU: follow-up; N/A: not applicable; SELNET: Sarcoma European and Latin American Network; SICO: Italian Society of Oncological Surgery; SIOPE: European Society of Paediatric Oncology; STS: soft tissue sarcomas; TH: telehealth; TO: tele-oncology; TR: telerehabilitation.

**Table 2 cancers-15-03700-t002:** Overview of articles with guidelines/practical recommendations for sarcoma patients during the COVID-19 pandemic.

First Author (Study)	Date	Study Type	Cancer Type	Remarks
Curigliano et al. [54]	2020	Review (ESMO practice guidelines for diagnosis, treatment and FU)	Various types	Telehealth for video consultations, triage, counseling, prescriptions, management via remote monitoring capabilities and wellness interventions (diet monitoring, physical activity, compliance, cognitive function and health risk assessment); Prioritizing care into three or four categories.
SSO [103]	2020	Resource for management options of sarcoma during COVID-19	Sarcoma	Νeoadjuvant CT for high-grade sarcomas for deferring surgical intervention if it can be safely delivered in an outpatient setting; Neoadjuvant imatinib in localized imatinib-sensitive GIST as a bridge therapy, even if a formal indication for neoadjuvant therapy does not exist.
Lenihan et al. (ICOS) [113]	2020	Review (guidelines)	Various types	Telemedicine and limited CV imaging for routine CV surveillance of cancer patients at low risk for CVD; Lower physical exposure for patients in semi-urgent need, such as uncontrolled blood pressure; Virtual cardio-oncology consultations if feasible during periods with high viral spread; Baseline ECG and QTc, as well as periodic assessment of cardiac biomarkers (TnI/TnT and NPs every 3– 6 weeks) in patients undergoing potentially cardiotoxic CT, such as anthracyclines; Routine home blood pressure monitoring in patients receiving TKIs; Prior RT: reduced cardiopulmonary reserve and increased susceptibility to a complicated COVID-19 infection; Antithrombotic prophylaxis in immobilized cardio-oncology patients with COVID-19; LMWHs constitute the first-choice antithrombotic therapy for this population.
Martin-Broto et al. (SELNET) [106]	2020	Review (guidelines)	Sarcoma	Telecommittees on a regular base; Surgery for localized high-grade conventional osteosarcoma, chondrosarcoma and skeletal EWS after neoadjuvant CT, as well as for other high-grade primary bone tumors; Neoadjuvant CT in localized osteosarcoma or localized EWS; Upfront CT in metastatic osteosarcoma or EWS; CT in recurrent advanced inoperable osteosarcoma or recurrent advanced EWS; RT should not be postponed for deep, neck, head STS or grade > 2 STS, which are over >5 cm, as well as RMS and extraskeletal EWS; TKIs for first, second, and third line in metastatic GIST should not be postponed; Perioperative CT for high-grade bone sarcoma, chondrosarcoma and beyond the third line in Ewing sarcoma, as well as the use of imatinib in GIST with <40% recurrence risk: low priority and could be delayed.
Burki et al. [107]	2020	Commentary	Sarcoma	The risk-benefit calculus remains challenging despite the patients’ categorization into priority groups based on their primary disease and the burden of their disease.
Janssens et al. (SIOPE radiation oncology group) [111]	2020	Review (guidelines)	Various pediatric tumors including sarcoma	CT can be employed to postpone RT in chemosensitive tumors, such as EWS and RMS; Reducing overall treatment time through similarly effective hypofractionated RT schedules and treatment gap corrections for EWS and RMS.
Cavaliere et al. (SICO) [105]	2021	Review (guidelines)	Various types including STS	A primary localized STS that needs surgery should be prioritized for surgery; Treatment postponement for 3–4 months in patients with benign biopsies, primary localized low-risk sarcomas, mesenchymal tumors of intermediate malignancy or GIST tumors which are stable on maintenance TKIs; Preoperative RT as a bridge therapy to postpone surgery if clinically feasible; Active observation or low-toxicity systemic treatment could be alternative options in patients with recurrent disease, such as those with indolent histologies.
Siavashpour et al. [112]	2021	Systematic review (guidelines) regarding RT	Various types including sarcoma	Preoperative RT can be employed for unresectable EWS; Neoadjuvant RT with a hypofractionated regimen may be employed for patients with large border-line resectable sarcomas; Brachytherapy in specialized centers (especially for RMS); Brachytherapy alone instead of 60−66 Gy/1.8−2 Gy/fr adjuvant external beam RT; Brachytherapy with high dose rate rather than low dose rate with iridium-192 wires.
Ben-Arye et al. (SIO) [114]	2021	Review, questionnaire, (online practice guidelines)	Various types	HP should address: (a) Patients’ skepticism toward the effectiveness of an online oncology treatment plan; (b) Ethical and legal issues regarding data protection and patients’ privacy; 2. HP should establish effective communication with patients.
ASCO [104]	2021	ASCO special report: A guide to cancer care delivery during the COVID-19 pandemic	Various types	Identification of appropriate patients for telemedicine (virtual check-in, e-visits, telephone) or a combination of telemedicine and in-person visits; Indications for telemedicine: FU, per os anticancer treatment adherence, (genetic) counseling, patient education, survivorship, palliative and supportive care, symptom monitoring, triage and quick patient assessment; Home infusion of CT is generally not recommended; Supportive care-related home infusion, such as hydration and anti-emetics, could be considered; Telemedicine practices may require extended hours to support patients’ needs.
Strauss et al. [109]	2021	Review (ESMO practice guidelines for diagnosis, treatment and FU)	Bone sarcomas	Multimodal CT and surgical resection for high-grade osteosarcomas; Heavy particle RT and IMRT for some patients with unresectable primary osteosarcomas; RFA and stereotactic RT: potential alternative local therapeutic modalities in osteosarcoma patients unfit for operation and those with small lung or bone metastatic disease; Denosumab: standard treatment for unresectable or metastatic giant cell tumors of bone, as well as preoperatively in some individualized complex cases; Lower frequency of FU visits for low-grade bone sarcomas (every 6 months for 2 years and, subsequently, annually).
Casali et al. [108]	2022	Review (ESMO practice guidelines for diagnosis, treatment and FU)	GIST	Adjuvant imatinib 400 mg/day without delay for at least 3 years in patients with locoregional GIST; No adjuvant therapy for patients with PDGFRA exon 18 D842V-mutated, NF1-related or SDH-negative GIST; Per os imatinib, sunitinib, regorafenib and repretinib: therapeutic choices for metastatic GIST; Larotrectinib or entrectinib for NTRK-rearranged GIST.

Overview of articles with guidelines or practical recommendations for sarcoma patients during the COVID-19 pandemic. ASCO: American Society of Clinical Oncology; CV: cardiovascular; CVD: cardiovascular disease; CT: chemotherapy; ECG: electrocardiogram; Ewing sarcoma: EWS; FU: follow-up; HP: healthcare professionals; ICOS: International Cardio-Oncology Society; IMRT: intensity-modulated radiation therapy; LMWH: low-molecular-weight heparin; N/A: not applicable; NF1: neurofibromatosis 1; NPs: natriuretic peptides NTRK: neurotrophic tyrosine receptor kinase; QTc: heart rate-corrected QT interval; rhabdomyosarcoma: RMS; RFA: radiofrequency ablation; RT: radiation therapy; SDH: succinate dehydrogenase; SELNET: Sarcoma European and Latin American Network; SICO: Italian Society of Oncological Surgery; SIO: Society for Integrative Oncology; SIOPE: European Society of Paediatric Oncology; SSO: Society of Surgical Oncology; STS: soft tissue sarcomas; TnI/TnT: cardiac troponins I or T; TO: tele-oncology; TKIs: Tyrosine Kinase Inhibitors.

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
