# Peer review of "Telemedicine in Care of Sarcoma Patients beyond the COVID-19 Pandemic: Challenges and Opportunities"

_cancers, 2023, doi:10.3390/cancers15143700_

Round 1
Reviewer 1 Report
- The study is a literature review that aims to analyze the use of telemedicine in different fields of sarcoma management.
- The "Abstract" is acceptable.
- The study design is correct, and paragraphs are clearly divided.
- However, the "Introduction" section is too long. The historical details could be described more briefly.
- I have a concern regarding the SEARCH end date, which is stated as May 31, 2023. It seems like a relatively short time for the analysis of data.
- In the flow-chart (Figure 1) and also in the text, it is not specified why 17 studies were excluded (from 165 to 148). This clarification is needed.
- Rows 166-167: It is not accurate to claim that all sarcomas are aggressive. Please rephrase this statement.
- From rows 166 to 176: I believe this paragraph should be placed in the "Introduction" section.
- It is recommended to use extended terms before corresponding acronyms (e.g., 199). Moreover, please use a consistent style when reporting data (e.g., p-value).
- The "Results" section is divided into clear subheadings, but it is too lengthy and contains some concept repetitions that the authors should avoid.
- The section on "Telemedicine guidelines for Telehealth and telemedicine practice and policy" contains too much information. The authors should avoid repeating information already presented in Table 2.
Author Response
#Reviewer 1:
The study is a literature review that aims to analyze the use of telemedicine in different fields of sarcoma management.
The "Abstract" is acceptable.
The study design is correct, and paragraphs are clearly divided.
However, the "Introduction" section is too long. The historical details could be described more briefly.
Response: We thank the reviewer for the positive feedback and we understand their concerns regarding the allocation of text to introductory details. We have, therefore, shortened the introduction drastically (pages 2-3). Particularly, the total word count of this section has been reduced by approximately 45% (-402 words).
I have a concern regarding the SEARCH end date, which is stated as May 31, 2023. It seems like a relatively short time for the analysis of data.
Response: We thank the reviewer for the demonstration of this concern. The original literature search, before tabulating and presenting the results, ended on the 22nd of May. However, we repeated the search on the 31st of May, one week before the final formatting and submission to ensure that we have not missed important findings published within the last week.
In the flow-chart (Figure 1) and also in the text, it is not specified why 17 studies were excluded (from 165 to 148). This clarification is needed.
Response: We thank the reviewer for bringing this to our attention. We have modified the flow diagram ( figure 1) adding further information regarding the reason why we excluded 17 (plus 7 studies) from the final reference list. These extra 7 studies were removed because they were considered irrelevant to the focus of the study on sarcoma patients according to the reviewers’ suggestions. Consequently, the final reference list consists of 141 references.
Page 4, section “2. Materials and Methods”, Figure 1: “Records excluded from the final reference list (concept repetitions or irrelevant to the focus of the study on sarcoma patients): -24 studies”
Rows 166-167: It is not accurate to claim that all sarcomas are aggressive. Please rephrase this statement. From rows 166 to 176: I believe this paragraph should be placed in the "Introduction" section.
Response: We thank the reviewer for this valuable suggestion. This misleading sentence has been modified and the whole paragraph has been placed in the introduction section, as requested.
Page 3, section 1.3. Telehealth in sarcoma care, lines 96-106
“Tele-oncology for sarcomas often requires a multidisciplinary approach for proper management.”
It is recommended to use extended terms before corresponding acronyms (e.g., 199). Moreover, please use a consistent style when reporting data (e.g., p-value).
Response: We thank the reviewer for these detailed comments. The review has been thoroughly checked and any abbreviations have been corrected. Furthermore, the reported data has been presented in a consistent style (“p-value:” instead of “p:”, “p”, or “p-”). An illustrative example can be found in the last paragraph of 3.1.1. Telemedicine and sarcoma diagnosis about a retrospective study regarding the delay of sarcomas before and after the pandemic (90 versus 103 days, respectively, p-value: 0.024), as reported in the original paper.
Page 7, line 164: “confidence interval (CI)”
Page 7, line 174: “soft tissue sarcomas (STS)”
Page 7, line 178 (previously line 199): “progression-free survival (PFS) and overall survival (OS)”
Page 9, line 301: “gastrointestinal stromal tumors (GIST)
Page 7, lines 159, 169, 170, 176, page 11, line 406 and page 13, line 474: “p-value:”.
The "Results" section is divided into clear subheadings, but it is too lengthy and contains some concept repetitions that the authors should avoid.
Response: We thank the reviewer for this suggestion. Subheadings have been properly modified. In particular, “3.1.1 Telemedicine and sarcoma diagnosis” has been substituted for “3.1.1 Telemedicine and sarcoma diagnosis during covid-19 pandemic”, “3.1.2 Telemedicine and surgery” has been used instead of “3.1.2 Telemedicine and surgery during covid-19” and “3.2. Telemedicine guidelines, practice and policy” has been chosen instead of “3.2. Telemedicine guidelines for Telehealth and telemedicine practice and policy”.
The section on "Telemedicine guidelines for Telehealth and telemedicine practice and policy" contains too much information. The authors should avoid repeating information already presented in Table 2.
Response: We thank the reviewer for the insightful feedback. This section (pages 13-14) has been changed and more than 200 words have been removed (decreased by ~25%) so that we can avoid overlapping with Table 2.

Reviewer 2 Report
Thank you for the opportunity to review this manuscript. The authors summarize literature related to telemedicine, COVID-19, and sarcoma and provide a comprehensive listing of many papers on this topic. Overall it is a very long paper that can be significantly shortened to emphasize the key points.
- The introduction should provide more sarcoma specific context. Some of the information early on the results describing the unique factors faced in sarcoma management can be moved as introduction.
- Many sections overlap with others. For example, sections “3.1.2. Telemedicine and surgery during covid-19” and “3.2. Telemedicine guidelines for Telehealth and telemedicine practice and policy” both provide long lists of various recommended guidelines that were developed during the pandemic. These sections can be condensed to highlight the overall message of the paper.
- Similarly, several aspects of the paper lose the focus on sarcoma. For example, section “3.1.5. Telemedicine and mental health during covid-19” does not contain any reference specifically to sarcoma. This section can be shortened and modified to be more sarcoma specific.
- A wider discussion of the impact and limitations of telemedicine on administering complicated sarcoma chemotherapy regimens should be included.
- Figure 2 seems to suggest that countries in the Northern Hemisphere are all high income and the lower hemisphere is all low income, with a somewhat problematic assignment of the challenges faced by each. This is a gross overgeneralization and should be modified.
Author Response
#Reviewer 2:
Thank you for the opportunity to review this manuscript. The authors summarize literature related to telemedicine, COVID-19, and sarcoma and provide a comprehensive listing of many papers on this topic. Overall it is a very long paper that can be significantly shortened to emphasize the key points.
- The introduction should provide more sarcoma specific context. Some of the information early on the results describing the unique factors faced in sarcoma management can be moved as introduction.
Response: We thank the reviewer for the positive feedback and we understand their concerns regarding the allocation of text to introductory details. We have therefore shortened the introduction drastically (pages 2-3). Particularly, the total word count of this section has been reduced by approximately 45% (-402 words).
Moreover, some general information regarding sarcoma has been placed in the introduction section instead of the section of results.
Page 3, section 1.3. Telehealth in sarcoma care, lines 96-106
Tele-oncology for sarcomas often requires a multidisciplinary approach for proper management. Sarcomas are a diverse group of tumors that can arise from bone or soft tissue, and they represent less than 1% of all adult cancers and 15% of all paediatric cancers. The prognosis for sarcoma patients is highly dependent on the stage and histologic subtype of the tumor, with 5-year survival rates ranging from less than 10% for advanced metastatic disease to over 80% for early-stage disease [9, 10]. One of the main concerns with tele-oncology for sarcomas is the potential for errors or delays in diagnosis, staging, and treatment planning. Sarcomas can be difficult to diagnose due to their rarity and diverse histologic subtypes. Consequently, misdiagnosis or delay in diagnosis can have serious consequences for patient outcomes [11].
- Many sections overlap with others. For example, sections “3.1.2. Telemedicine and surgery during covid-19” and “3.2. Telemedicine guidelines for Telehealth and telemedicine practice and policy” both provide long lists of various recommended guidelines that were developed during the pandemic. These sections can be condensed to highlight the overall message of the paper.
Response: We understand the concerns that the reviewer raised. We have modified the section to focus only on the effect of the pandemic on the surgical management of cancer patients, a fact which necessitated the prompt reaction of oncologic societies worldwide with an extensive consensus among sarcoma experts. These guidelines will be further illustrated in section 3.2. Telemedicine guidelines, practice and policy. We then reported some studies regarding a new triage model in Quebec as documented by Olshinka et al and the reported results of the pandemic among musculoskeletal oncology surgeons. Section 3.2 was shortened as many guidelines overlap with the ones mentioned in Table 2.
- Similarly, several aspects of the paper lose focus on sarcoma. For example, section “3.1.5. Telemedicine and mental health during covid-19” does not contain any reference specifically to sarcoma. This section can be shortened and modified to be more sarcoma specific.
Response: We thank the reviewer for this comment. Studies referring to the psychological consequences of the pandemic on the general population have been omitted. We have reported a single study that focuses entirely on sarcoma patients and the effects of the pandemic on their mental health. We wanted to highlight the effect of telemedicine on mental health and, thus, we included a meta-analysis that focuses on breast cancer patients.
Pages 11-12, Section: 3.1.5. Telemedicine and mental health during covid-19, lines 397-410:
Literature focuses on the mental burden of cancer patients as a vulnerable population and rarely categorizes them into specific cancer types.
Younger et al have investigated the psychological impact of the pandemic on sarcoma patients. In this survey of 350 patients with sarcoma, an increase in loneliness, feelings of worry and insomnia were reported, especially among participants in palliative care or without a clear treatment intent. 259 out of 350 patients (74%) indicated a preference for telehealth appointments in the future compared with 78 patients (22%) that would prefer only face-to-face appointments. The treatment intent and the resilient coping level were significantly related to the preference for only in-person appointments (p-value: 0.047 and 0.024, respectively). In particular, patients with unknown treatment intent or a low resilient coping score were most likely to prefer face-to-face contact. A decrease in all aspects of all Health-Related QoL (HRQoL) concerning physical, emotional and social functioning compared to pre-pandemic was also documented [14].
- A wider discussion of the impact and limitations of telemedicine on administering complicated sarcoma chemotherapy regimens should be included.
Response: We thank the reviewer for the insightful feedback. We have meticulously searched the relevant literature and we included further information about the impact and limitations of telemedicine on administering complicated sarcoma chemotherapy regimens. More specifically, a new paragraph has been added regarding further limitations of telemedicine in the section of the discussion (4.2).
Page 20, section: “4.1. Benefits of telemedicine in cancer care”, lines: 615-616: Telehealth has the potential to reduce the infection risk, as well as the risk of chemotherapy-induced myelosuppression in high-risk patients [115].
Page 20, section: “4.1. Benefits of telemedicine in cancer care”, lines: 637-639: Finally, telemedicine can be beneficial to patients in long-term follow-up in financial terms, given that it can alleviate travel and lodging expenses and reduce missed work days [115].
Page 21, section “4.2 Challenges of telemedicine”, lines: 671-677:
However, sarcoma patients often receive complex treatment regimens that require admission and hospitalization. The pandemic caused significant interruptions to these patients’ therapeutic plans, but telehealth can not effectively address these emerging issues. A striking example is that many patients with complex radiation and chemotherapy schedules were required to either modify their regimen or delay their treatment. Another considerable limitation of this group of patients was the difficulty in obtaining laboratory or radiologic imaging studies [130].
- Figure 2 seems to suggest that countries in the Northern Hemisphere are all high-income and the lower hemisphere is all low income, with a somewhat problematic assignment of the challenges faced by each. This is a gross overgeneralization and should be modified.
Response: We thank the reviewer for bringing this issue to our attention, we would certainly not wish to promote a false understanding of income and health inequalities between the global north and south. We have interchangeably replaced the terms high and middle-and-low-income countries in the text, as well as on the superior and inferior part of the figure to highlight that income distribution is variable across both hemispheres, while all of the countries are vulnerable to regional and global challenges ranging from armed conflict to climate change.
Page 21, “Section 4.2. Challenges of telemedicine”, line 662: “between the high and middle-and-low-income countries”
Page 22, “Section 4.2. Challenges of telemedicine”, lines 688-689 and Figure 2. “Figure 2. Challenges regarding telemedicine. Societal, regulatory and clinical factors pose barriers to remote sarcoma care being compounded by structural inequalities across high and low-and-middle-income countries and emerging political, financial and environmental crises.”

Round 2
Reviewer 2 Report
Thank you for addressing the comments.